# Single Seed Identification in Three *Medicago* Species via Multispectral Imaging Combined with Stacking Ensemble Learning

**DOI:** 10.3390/s22197521

**Published:** 2022-10-04

**Authors:** Zhicheng Jia, Ming Sun, Chengming Ou, Shoujiang Sun, Chunli Mao, Liu Hong, Juan Wang, Manli Li, Shangang Jia, Peisheng Mao

**Affiliations:** College of Grassland Science and Technology, China Agricultural University, Beijing 100193, China

**Keywords:** *M. falcata*, *M. varia*, *M. sativa*, seed identification, stacking ensemble learning, multispectral imaging

## Abstract

Multispectral imaging (MSI) has become a new fast and non-destructive detection method in seed identification. Previous research has usually focused on single models in MSI data analysis, which always employed all features and increased the risk to efficiency and that of system cost. In this study, we developed a stacking ensemble learning (SEL) model for successfully identifying a single seed of sickle alfalfa (*Medicago falcata*), hybrid alfalfa (*M. varia*), and alfalfa (*M. sativa*). SEL adopted a three-layer structure, i.e., level 0 with principal component analysis (PCA), linear discriminant analysis (LDA), and quadratic discriminant analysis (QDA) as models of dimensionality reduction and feature extraction (DRFE); level 1 with support vector machine (SVM), multiple logistic regression (MLR), generalized linear models with elastic net regularization (GLMNET), and eXtreme Gradient Boosting (XGBoost) as basic learners; and level 3 with XGBoost as meta-learner. We confirmed that the values of overall accuracy, kappa, precision, sensitivity, specificity, and sensitivity in the SEL model were all significantly higher than those in basic models alone, based on both spectral features and a combination of morphological and spectral features. Furthermore, we also developed a feature filtering process and successfully selected 5 optimal features out of 33 ones, which corresponded to the contents of chlorophyll, anthocyanin, fat, and moisture in seeds. Our SEL model in MSI data analysis provided a new way for seed identification, and the feature filter process potentially could be used widely for development of a low-cost and narrow-channel sensor.

## 1. Introduction

The cultivated species in the *Medicago* genus, including sickle alfalfa (*M. falcata*), hybrid alfalfa (*M. varia*), and alfalfa (*M. sativa*), are widely distributed in arid and semi-arid regions. They were used as high-quality forage crops, ecologically protected herbages, and products for various purposes (e.g., energy substances, food, and nutraceuticals) around the world [1,2,3].

With increasing market demand, new species or varieties have been introduced to satisfy the requirements for *Medicago*. *Medicago* species have differences in production characteristics, regional suitability, adversity tolerance, and economic value [4]. Thus, distinguishing seeds of *Medicago* species for maintaining seed purity and genuineness are beneficial for the interests of seed companies, farmers, and dealers, resulting in the healthy development of the seed industry. However, the seed morphologies of sickle alfalfa, hybrid alfalfa, and alfalfa are similar, and so professionally trained personnel are required to detect them, which is time-consuming and mostly destructive [5]. In addition, molecular markers are also crucial techniques for identifying species and varieties with high accuracy [6], but this method is destructive and cannot allow real-time detection and sorting. Therefore, it is important to develop a rapid, non-destructive, and high-throughput method for *Medicago* seed identification, which is also a focus for seed testing research [7].

Multispectral imaging (MSI), a non-destructive technique that combines computer vision and spectroscopy, can provide data of texture, color, shape, size, and chemical composition [8,9]. The principle of MSI is to analyze the different physical structures and the chemical composition of objects by detecting whether these objects can absorb or reflect specific wavelengths [10]. Currently, this technique is widely used in the food, pharmaceutical, and agricultural fields. In seeds, it is mainly applied to distinguishing varieties and detecting seed health and seed viability. For example, MSI has been successfully applied to distinguishing alfalfa from sweet clover (*Melilotus albus*) seeds [10] and detecting similarity between transgenic rice (*Oryza sativa*) and non-transgenic rice seeds [11] as well as different varieties of pepper (*Capsicum annuum*) seeds [12]. Despite the desirable classification results obtained by applying MSI in analyzing and detecting the quality of crop seeds, it is still challenging to turn these laboratory-based techniques into more practical applications, owing to the high cost. Thus, a low-cost imaging system capable of obtaining spectral features at specific key wavelengths and viable classification models to monitor crop seed quality are urgently needed [13].

Machine learning (ML) has been used as a powerful algorithm to deal with “big data” [14,15] and can automatically learn data patterns and optimize model parameters. Meanwhile, ML has been widely applied in processing MSI data [11]. ML includes traditional algorithms, for example, support vector machine (SVM) [16] and linear discriminant analysis (LDA) [17], but also deep learning (DL), such as convolutional neural network (CNN) [18] and generative adversarial network (GAN) [19]. However, DL is not suitable for small number samples, because large number samples and many parameters need to be tuned.

The traditional ML algorithm requires the sample probability distribution of the test dataset to be consistent with that of the training dataset. In this way, the specific distribution that matches the actual situation can be determined as a classifier among many hypothetical functions. However, the wide distribution and diverse production measures of *Medicago* cause significant individual differences in seeds, and so data distribution of the actual test set is uncertain, which leads to the weak generalization ability of a single classifier. To overcome this problem, ensemble learning methods are proposed to accomplish a high-precision classification task by ensembling multiple weak classifiers into a robust classifier. The most commonly used ensemble learning methods include parallelized ensemble of bagging [20], sequential integration of boosting [21], and multilayer classification stacking [22]. Stacking ensemble learning (SEL) could significantly improve the predictive force of models compared with bagging and boosting ensemble learning. It has been widely applied in education, medicine, and social sciences [23,24] but rarely in agriculture [25]. In particular, *Medicago* seed discrimination via MSI combined with a SEL model has not been reported.

Thus, the aims of this study are (1) to propose a SEL model for discriminating single seeds of three *Medicago* species and benchmark the performance of SEL and basic models and (2) select the potential optimal features for distinguishing single seeds of three *Medicago* species and explain possible differences in substance content.

## 2. Materials and Methods

### 2.1. Materials

Sickle alfalfa seeds were purchased from Beijing Crawford Ecological Technology Co., Ltd., Beijing, China; hybrid alfalfa seeds were provided by the Forage Seed Laboratory, China Agricultural University, and alfalfa seeds were purchased from Gansu Jiuquan Daye Seed Co., Ltd., Jiuquan, China. Images of the seeds are displayed in Figure 1.

For each species, 200 seeds were selected for classification experiments within each species, among which 70% were randomly selected as a training set and 30% as the testing set. The seed information of each species, including origin, moisture content, and storage conditions, is listed in Table 1.

### 2.2. Multispectral Imaging System

MSI data of seeds were acquired using the VideometerLab4^TM^ (Videometer A/S, Hørsholm, Denmark) MSI system. Before capturing multispectral images, the system was fully calibrated radiometrically and geometrically by using three successive plates: white, dark, and dotted, followed by a light setup calibration. The samples were illuminated by high-power light-emitting diodes (LEDs) at the edge of the sphere, ranging from ultraviolet (UV) to near-infrared (NIR) at 19 specific wavelengths: 365, 405, 430, 450, 470, 490, 515, 540, 570, 590, 630, 645, 660, 690, 780, 850, 880, 940, and 970 nm. The LEDs strobed sequentially in approximately 5 s of scanning time and produced monochrome images of 19 different wavelengths. These images had a high spatial resolution of approximately 40 µm/pixel, consisting of 2192 × 2192 pixels. RGB images were also captured by the same sensor, where each individual pixel was associated with a value in the red, green, and blue channels.

### 2.3. Multispectral Image Analysis

After the images were acquired, each seed was segmented from the background as a region of interest (ROI) using VideometerLab 3.14 software. Then, all seeds were collected in a blob database, and different features of the seeds could be extracted, such as morphological (from RGB images) and spectral features (from multispectral images) (Appendix A). The extracted spectral features of the seeds represent the average intensity of reflected light at each single wavelength calculated from all seed pixels in the image. Thus, the average reflected spectrum of any seed in the image is represented by 19 values that are calculated by averaging the intensity of pixels in 19 wavelength bands of 365 nm to 970 nm in the ROI of the seed. In total, average spectra of 600 samples were achieved in this study. The spectral and morphological features of the seeds were then all collected in a matrix (X) that was associated with their corresponding species data (Y).

### 2.4. SEL Model

We designed a SEL model containing three levels: level 0 for dimensionality reduction and feature extraction (DRFE) models, level 1 for basic models, and level 2 for a meta-learner. DRFE models (level 0) relied on different kinds of DRFE algorithms to learn from the original dataset to produce a fused feature dataset (level 1), while level 1 models produced the meta-feature fused dataset for level 2, and the final meta-model (level 2) produced the final result (Figure 1).

To avoid overfitting and ensure participation of all original datasets in the training of generating new meta-feature datasets, nested resampling was used in the training of a single basic model and the performance evaluation of the model. Nested resampling includes internal k-fold cross-validation and external k-fold cross-validation (internal k-fold and external k-fold to 10 and 3, respectively). The training process of the basic model and the structure of the entire superimposed model are shown in Figure 2.

The general steps of the SEL were as follows.

(1) The original dataset was divided into a training set and a test set.

(2) For each DRFE model at level 0, nested resampling was used for training (external k-fold resampling and internal k-fold resampling) and fusing the model output dataset in each fold into a new feature set, and then the new feature set was used for training the model in the next level.

(3) Each basic model at level 1 was trained by nested resampling using the union new dataset of level 0, and the output data of each basic model were fused into a new meta-dataset. This was used for the training of the meta-learner.

(4) The new meta-dataset was used to train the meta-learner at level 2 and output the final prediction results.

In summary, DRFE reduced the dimensionality of the features in the training set, and we fused the results of different dimensionality reduction algorithms together to form a new feature dataset (a concatenation of the first two dimensions of each dimensionality reduction algorithm), which largely reduced the ‘digital dimensionality disaster’. The new feature dataset was then used for the training of the second layer model, followed by merging the results of each model prediction into a new feature set as the meta-learner input.

### 2.5. Selection of DRFE Models, Basic Models, and the Meta-Learner

For DRFE models at level 0, principal component analysis (PCA), LDA, and quadratic discriminant analysis (QDA) models were selected. PCA is generally used to reduce the dimensionality of the data through a mathematical technique by an orthogonal transformation of the initial dataset into a new set of uncorrelated variables, the so-called principal components (PCs), where the first PC has the highest variance, the second PC has the second-highest variance, and so on. Thus, key information and a potential data structure of high-dimensional data can be provided by PCs. LDA, a classical ML algorithm, calculates the optimal transformation (projection) by simultaneously minimizing the within-class distance and maximizing the between-class distance and results in maximum discrimination. QDA, a variant of LDA, separates groups nonlinearly by estimating a covariance matrix for each of them. In brief, DRFE models could reduce data dimensionality and extract the critical information in raw data at level 0.

For basic models at level 1, SVM, multivariate logistic regression (MLR), eXtreme Gradient Boosting (XGBoost), and generalized linear models with elastic net regularization (GLMNET) were employed. SVM, a well-known kernel method, has been effectively used for multivariate function estimation or nonlinear classification by finding the optimal hyperplane to achieve segmentation of high-dimensional data [26,27]. MLR is used to predict the probability of category placement or category affiliation with a dependent variable based on multiple independent variables. The independent variables can be dichotomous (i.e., binary) or continuous (i.e., interval or proportional). It is a simple extension of binary logistic regression that uses maximum likelihood estimation to assess the probability of category affiliation for classification purposes [28]. GLMNET uses the elastic network technique. Elastic network is a regularization technique, combining least absolute shrinkage and selection operator (LASSO) and ridge regression. LASSO utilizes L1 regularization as a penalty method, while ridge regression utilizes L2 regularization [29]. XGBoost performs a second-order Taylor expansion on the objective function and uses the second-order derivative to speed up the convergence of the model during training [30]. In addition, a regularization term is added to the objective function to control the complexity of the tree, thereby generating a relatively simple model and preventing overfitting [31].

Finally, XGBoost was applied as the meta-learner at level 3, because XGBoost could correct the bias of multiple algorithms on the training set. Furthermore, the hyperparameters of each basic model were tuned by the random-search method. The optimized ranges of selected and tuned hyperparameters of basic models and the meta model are listed in Table 2, and those hyperparameters not mentioned in this table were assigned as default values. All algorithms and tuning parameters were implemented through the ‘mlr3’ package in R (version 4.05) [32].

### 2.6. Evaluation Metrics and Model Interpretation

We selected accuracy, kappa, precision, sensitivity, and specificity as evaluation metrics for model performance. Their equations are listed as follows.
(1)$$\mathrm{Accuracy}\;=\frac{TP+TN}{TP+TN+FP+FN}$$
(2)$$\mathrm{Kappa}\;=2\times \frac{TP\times TN-FP\times FN}{TP\times FN+TP\times FP+2\times TP\times TN+FN^{2}+FN\times TN+FP^{2}+FP\times TN}$$
(3)$$\mathrm{Precision}\;=\frac{TP}{TP+FP}$$
(4)$$\mathrm{Sensitivity}\;=\frac{TP}{TP+FN}$$
(5)$$\mathrm{Specificity}\;=\frac{TN}{TN+FP}$$
where *TP*, *FP*, *TN*, and *FN* represent true positive, false positive, true negative, and false negative, respectively.

For model interpretation, XGBoost, MLR, and GLMNET algorithms had their own way of quantifying feature importance. XGBoost calculated which feature was selected as the splitting point based on the gain of the structure score, and the importance of a feature was the sum of its occurrences in all trees. SVM and SEL were interpreted based on a permutation feature importance algorithm [33] which was independent of the model, and the feature importance measure was calculated by calculating the prediction error of models when the prediction errors of features increased after permutations. A feature is ‘important’ if its value increases the error of the model, because the model relies on this feature a great deal. If the model error remains the same, the feature is ‘unimportant’, because the model ignores the feature in its prediction. The key formulation was listed as follows.
(6)$$e^{orig\;}\;L(y,f(X))$$
(7)$$e^{perm\;}=L\left(Y,f\left(X^{perm\;}\right)\right)$$
(8)$$FI^{j}=e^{perm\;}/e^{orig\;}$$
where *f* represents the model; *X* indicates the feature matrix; *y* means the target vector; *L*(*y, f*) denotes the error measure; *e^orig^* represents the error estimate of the original model (6); and X*^perm^* is generated by replacing feature *j* in the data *X* after breaking the association between feature *j* and the true outcome *y*; *e^perm^* denotes the error estimate of the model features after replacing features (7). *FI^j^* denotes the importance of feature variable *j* (8). For example, a SVM model was first trained based on the training set, and then SVM was tested on the test set to attain the core of the model metrics ‘ce’. After that, a feature was randomly replaced on the test set, and the new ‘ce’ score was obtained after using SVM for prediction. A comparison was made with the metric obtained in step 2, and if the difference was large, this indicated that the feature was more important. Such steps were followed until all features were calculated.

Finally, feature importance scores of all models were scaled to the range of 0 to 100 for better comparison of features on different models.

### 2.7. Optimal Features Selection

We adopted filter methods that assigned an importance value to each feature. Based on these values, the features could be ranked, and a feature subset could be selected. First, we calculated the feature importance of each model (SEL as an entirety model) based on all features. Then, we selected the feature importance of the top three ranked models after the scores of feature importance over 50 were selected. Finally, the intersection of the important features of the top three performance models was employed as the optimal features.

## 3. Results

### 3.1. Morphological Features of Seeds in Three Medicago Species

There were 14 morphological features extracted from the multispectral images, which showed differences among seeds of three *Medicago* species in terms of the mean value of binary features, shape features, and color features. For binary features, the area of hybrid alfalfa seed was the largest, while sickle alfalfa seed was the smallest. Additionally, no significant differences were observed among seeds of the three *Medicago* species in width and length (Appendix A).

For shape features, the highest value of Beta shape b and the lowest value of compactness circle were found in sickle alfalfa seeds, and hybrid alfalfa seeds had the lowest values of Beta shape a and Beta shape b. In contrast, the three species showed no differences in compactness ellipse, vertical orientation, and vertical skewness (Appendix A).

For color features, the seeds of the three species were different, and, especially in alfalfa, the mean value of hue was the highest, while CIELab A* was the lowest. The values of hue, CIELab B*, and saturation were the lowest in sickle alfalfa. Nevertheless, hybrid alfalfa had the lowest value of CIELab L* and the highest value of saturation (Appendix A).

### 3.2. Spectroscopic Analysis of Seeds in Three Medicago Species

Overall, a similar trend was shown for mean spectral reflectance in seeds of three *Medicago* species. We found that the longer the wavelength, the higher was the reflectance (Figure 3a). Especially in the spectral ranges of 365 nm to 540 nm and 645 nm to 690 nm, the highest reflectance was present in sickle alfalfa seeds. In the NIR region (780–970 nm), the spectral reflectance of alfalfa seeds was significantly higher than the others (*p* < 0.05) (Appendix A). Additionally, the variation coefficients of spectral reflectance for seeds were higher in the UV (365 nm) and visible light (VL) regions (405–690 nm) than those in other wavelengths. The variation coefficient of spectral reflectance for hybrid alfalfa seeds was the largest in the wavelength range of 365 nm to 470 nm, while the lowest was in the wavelength range of 470 nm to 645 nm. There were similar spectral reflectance variations coefficient in UV, VL, and NIR regions for sickle alfalfa and alfalfa (Figure 3b).

The results of PCA showed that the first two PCs for morphological features explained 47.2% of the total variance among seeds, with PC1 of 26.4% and PC2 of 20.8%, respectively (Figure 4a). For the spectral features, the first two PCs explained 59.5% and 28.1% of the variances, respectively (Figure 4b). In addition, based on combined morphological and spectral features, the first two PCs (PC1 of 36.6% and PC2 of 15.8%) explained a total of 52.4% of the original variance (Figure 4c). However, although the first two PCs had explained more variances based on spectral features, no significant differences were observed among seeds of the three *Medicago* species.

The results of LDA showed that the first two LDs explained 81.28% and 18.72% of the variance for morphological features, respectively. However, the seeds of the three *Medicago* species could not be separated from each other either by LD1 or LD2, as there was less variance among the seeds (Figure 5a). For spectral features, the first two LDs explained a total of 100% of the variance, as seeds of alfalfa could be distinguished entirely from sickle alfalfa and hybrid alfalfa in LD1, and seeds of hybrid alfalfa could be significantly distinguished from sickle alfalfa and alfalfa by LD2 (Figure 5b). For combined spectral and morphological features, the first two LDs explained a total of 100% of the variance, and LD1 and LD2 explained 90.77% and 9.23% of the variance, respectively. Moreover, sickle alfalfa seeds could be separated by LD1 from seeds of hybrid alfalfa and alfalfa (Figure 5c). In conclusion, the results of PCA and LDA showed that morphological features cannot be taken as input data alone for the models.

### 3.3. SEL Model for Seed Discrimination

For spectral data, the SEL, SVM, MLR, and GLMNET worked well, and their values of overall accuracy, kappa, sensitivity, specificity, and precision were all above 0.90, while SEL attained the highest value of 1.00 (Figure 6a). Similarly, SEL was the best for all parameters in the combined morphological and spectroscopic dataset (Figure 6b). On the training set, testing set, and cross-validation set, the indicator of hit rate presented the highest values of accuracy, sensitivity, specificity, and precision in all species. In particular, seeds of *M. sativa* obtained the highest value of 1.00 in terms of accuracy, sensitivity, specificity, and precision (Table 3).

For the combination of spectral and morphological data, SEL obtained the highest values of overall accuracy, kappa, sensitivity, specificity, and precision, followed by MLR, SVM, GLMNET, and XGBoost (Figure 5b). SEL showed the highest overall accuracy of 0.94 on testing data. The highest hit rate of sickle alfalfa seeds was 100%, with sensitivity greater than 0.98. The values of individual accuracy and sensitivity were the lowest for hybrid alfalfa, with at least 13% of seeds misclassified as alfalfa. (Table 4).

To evaluate the working patterns of SEL and the four basic models, the feature importance scores were calculated for all features, including morphological and spectral features. For MLR, there were 15 features with importance scores of over 50. Moreover, VL wavelengths (570 and 590 nm) were more important (Figure 7a). There were four features with XGBoost’s feature score of greater than 50, with 970 and 365 nm being the two most important features (Figure 7b). SEL had 15 features with importance scores of greater than 50, and the NIR region (880 and 940 nm) was among the most important features (Figure 7c). For SVM, 365 and 970 nm were important features (Figure 7d). For GLMNT, the importance scores of at least 17 features were 0, and only 970 and 780 nm were considered as important features with scores of greater than 50, while the other feature scores were all less than 50 (Figure 7e).

### 3.4. Optimal Feature Selection for Identifying Medicago Species Seeds

The intersection of more important features (score > 50) of SEL, SVM, and MLR was considered as the optimal features, which included five features: 365, 515, 570, 880, and 970 nm (Figure 8a). With these five features as input data, the accuracy of SEL, SVM, and MLR on cross-validation sets was above 0.90, and SEL had the highest value of 0.97 (Figure 8b). Among the five features on SEL, 970 nm was the most important, followed by 880, 365, 515, and 570 nm (Figure 8c).

Moreover, other performance metrics and model effects of SEL were evaluated based on these five wavelengths. The results showed that the values of kappa, sensitivity, specificity, and accuracy ranged from 0.90 to 1.00 (Figure 8d). There was an opposite trend of influence for seeds of sickle alfalfa and hybrid alfalfa at 365 nm. On the VL region (515 and 570 nm), the higher the spectral reflectance at 515 nm was, the more possible hybrid alfalfa seeds were. In contrast, at 570 nm, the lower the spectral reflectance was, the more possible hybrid alfalfa seeds were. In the NIR region (880 and 970 nm), sickle and hybrid alfalfa seeds showed opposite trends in model effects, and hybrid alfalfa seeds maintained similar trends to alfalfa seeds (Figure 8e).

Additionally, the results of the density of spectral reflection distribution showed differently distributed peaks among seeds of sickle alfalfa, hybrid alfalfa, and alfalfa at 365 nm, while the peak value of hybrid alfalfa was lower than those in the other two ones. In the VL region (515 and 570 nm), there were different peaks in reflectance for the three species seeds in *Medicago*. In addition, in the NIR region, the peak values of the three species seeds had different distributions, with the largest peak values of hybrid alfalfa at 880 nm and alfalfa at 970 nm (Appendix A).

## 4. Discussion

According to the International Seed Federation (ISF), worldwide seed trade reached 5.68 million metric tons in 2018, with a market size of $138.1 billion (https://www.worldseed.org/resources/seed-statistics/) (accessed on 5 March 2022). However, adulteration of seeds is also a great concern for growers and consumers and sometimes causes huge economic losses. Therefore, it is of great significance to identify seed varieties and grades [34]. With the development of spectroscopy and computer technologies, spectral imaging technology can obtain different dimensional information by seed image information combined with spectral information. Many companies and organizations try to explore these new technologies for seed testing. Previous studies showed that morphological and spectral features of different seeds were related to species, physiological status, and material content, which could be used for seed classification [35].

MSI provides rapid capture of morphological and spectral features of single seeds and is used to distinguish morphologically similar seeds among species or varieties. Yang et al. [36] reported that morphological features could not distinguish 12 varieties of alfalfa seed. In this study, the exploratory quantitative analysis results of PCA and LDA clearly showed that the intra-species variation in morphological features of three *Medicago* species seeds were greater than the inter-species variation. In addition, the first two PCs explained only 47.2% of the variance, which suggested that there was no significant difference in morphological features among the three species. There were several possible reasons for this result. Firstly, the seeds from the same genus, species, or variety might have similar shapes, because they might be harvested in the same region with similar climatic environments and with similar agricultural production methods [37]. Secondly, perennial species in *Medicago* constitute a kind of cross-pollination plant with relatively large variation within species [38]. Therefore, PCA and LDA could not detect different seeds among species when within-species variation was close to or greater than among-species variation. Spectral information varied among species and varieties, such as wheat (*Triticum aestivum*) and maize (*Zea mays*) [34,37]. Similar results were observed in this study, i.e., that the spectral reflection varied significantly among different *Medicago* species seeds. Although the first two PCs explained a total of 87.6% of the variance for PCA, there were no differences among the three species, indicating that there was still a large intra-species variation. Obviously, these variations were mainly in the UV and VL regions. In contrast, the seeds of the three species could be distinguished by LDA (supervised learning), whose principle is to achieve separation by maximizing the variance between groups. Yu et al. [39] also found that LDA was superior to PCA in the exploratory quantitative classification of 18 okra (*Abelmoschus esculentus*) varieties’ seeds.

Most previous research has focused on the combination of a single ML model in MSI analysis [40,41]. In this study, a new SEL model, containing three-level models, was superior to all basic models according to the metrics of overall accuracy, kappa, sensitivity, specificity, and precision. For spectral data, the individual classification metrics of the three species in the confusion matrix of SEL all performed the best. For the combination of morphological and spectral data, the classification performance of the four basic models decreased significantly (accuracy < 0.90), while the classification accuracy of SEL was still higher than 0.94, and the values of sensitivity, accuracy, specificity, and kappa were all around 0.90. This proved that the generalization ability and robustness of SEL were much better. Additionally, we observed that all models performed better on the spectral dataset than on the morphological-spectral dataset, suggesting that spectral data included the main differences between the three species’ alfalfa seeds, which also supported the results of the feature filters (i.e., the important features were all spectral features). However, it should be noted that, although SEL had a very high prediction performance, the training time was time-consuming, and we should optimize the training time by adjusting the cross-fold number of SEL for specific tasks. In addition, there were differences between SEL and basic models. In total, 15 features (score > 50) in SEL or MLR were used to carry out the discriminating tasks, but important features in the NIR band were selected for SEL, while the ones in the VL band were selected for MLR. Furthermore, compared with SEL or MLR, only a few features (scores above 50) could be used for seed classification in XGBoost, SVM, and GLMNET. Thus, the number and performed spectral range of important features were the key factors causing the differences in their classification performance among models. It was also reported that the LDA and SVM models worked under different input features [21,36]. For combined morphological and spectral data, LDA was dominated by morphological features, and SVM was dominated by spectral features.

Feature information redundancy generally increases the complexity of model training and reduces the performance of the model. Additionally, multi-channel band acquisition would increase the cost of the system and limit the practical applications [8,42]. The development of a sensor with a low channel band could reduce the production cost and improve computational efficiency [43]. Weng et al. [13] developed a low-cost, narrow-band MSI system for rapid detection of rice false smut (RFS) and found that spectral features at 460, 520, 660, 740, 850, and 940 nm were well associated with RFS, and the accuracy of models was over 90%. In this study, we employed a filtering method which selected the intersection of the important features within models of the top three accuracies, successfully reduced the number of features from 33 to 5, and improved the model’s performance. Five wavelengths, including 365, 515, 570, 870, and 970 nm, were more important features for the discrimination of *Medicago* seeds, which potentially indicated internal physiological and seed coat differences among seeds of different *Medicago* species. The 365 nm wavelength was generally associated with chlorophyll [38], and the model effect showed that this difference was primarily between seeds of sickle alfalfa and hybrid alfalfa at 365 nm. Carotenoid and anthocyanin were generally relevant at 515 and 570 nm [44]. Model effects at the 515 and 570 nm wavelengths showed differences among the seeds of the three *Medicago* species, while sickle alfalfa and alfalfa seeds showed similar levels of these two wavelengths. The wavelength of 870 nm was generally associated with fat [8], which is an important nutrient in seeds, and its model effect showed that that difference mainly lay in the seeds of sickle alfalfa and alfalfa, with hybrid alfalfa similar to alfalfa. The wavelength of 970 nm was generally related to moisture [45], and its effects on alfalfa and sickle alfalfa were opposite, with alfalfa similar to hybrid alfalfa, which may have been due to the higher moisture content of sickle alfalfa than of those in the others. In short, there was a closer relationship between alfalfa and hybrid alfalfa seeds at the bands of UV (365 nm) and NIR (870 nm), while sickle alfalfa and alfalfa seeds were similar at the VL region (515 and 570 nm).

## 5. Conclusions

This study developed a SEL model combined with multispectral technology to identify and distinguish single seeds of three *Medicago* species. Meanwhile, the SEL model showed a more powerful generalization ability and robustness than the basic learners in the metrics of model evaluation. The number of features was successfully reduced from 33 to 5 by selecting the intersection of more important features in models of the top three ranked performances, and SEL could still maintain and improve performance. These results provided a powerful model for discriminating *Medicago* species seeds and a theoretical basis for the development of low-cost and high-efficiency sensors. Meanwhile, the association of MSI features, phenotypes, and internal components should be focused on in the future, and the development of new, robust, efficient, and low-cost sensor models is needed.

## Figures and Tables

**Figure 1 sensors-22-07521-f001:**
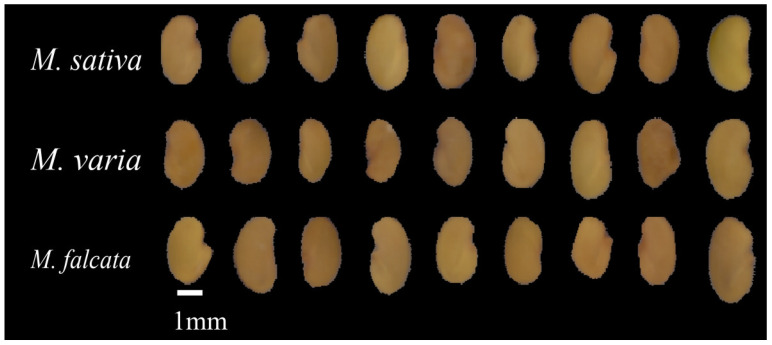
Seed image of alfalfa (*M. sativa*), hybrid alfalfa (*M. varia*), and sickle alfalfa (*M. falcata*).

**Figure 2 sensors-22-07521-f002:**
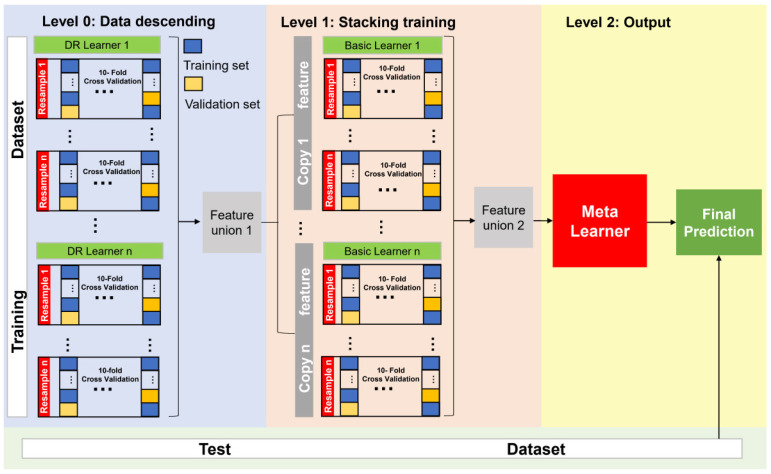
Structure of the SEL Model.

**Figure 3 sensors-22-07521-f003:**
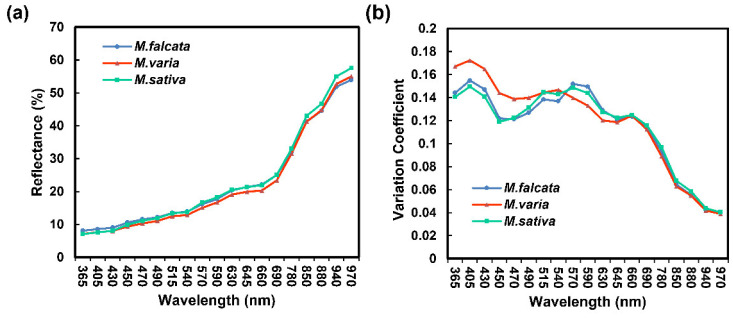
Spectral information of three *Medicago* species seeds. (**a**) Average spectral reflectance of 19 wavelengths. (**b**) Variation coefficient of 19 wavelengths.

**Figure 4 sensors-22-07521-f004:**
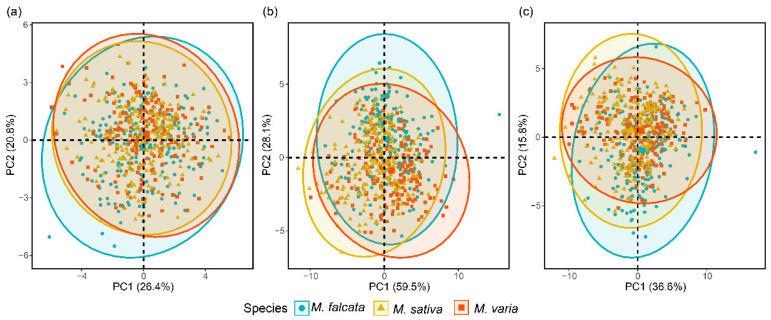
PCA score plot of three *Medicago* species using values of morphology (**a**), spectra (**b**), and combination of morphological and spectral features (**c**).

**Figure 5 sensors-22-07521-f005:**
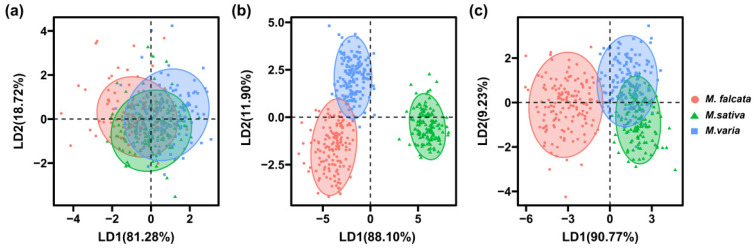
Two-dimensional biplot of LDA scores distinguishing seeds of three *Medicago* species based on morphological features (**a**), spectral features (**b**), and combined spectral and morphological features (**c**).

**Figure 6 sensors-22-07521-f006:**
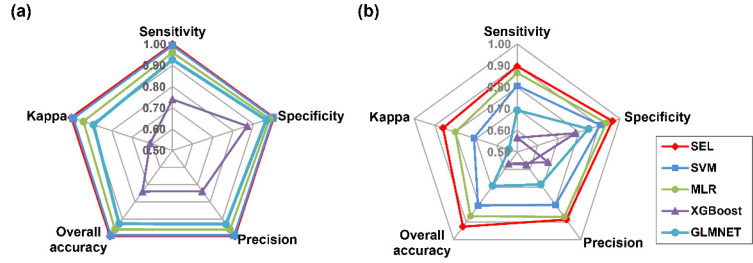
Radar plot of model classification performances on test data by overall accuracy, precision, specificity, sensitivity, and kappa, based on spectral features (**a**) and combination of spectral and morphological features (**b**).

**Figure 7 sensors-22-07521-f007:**
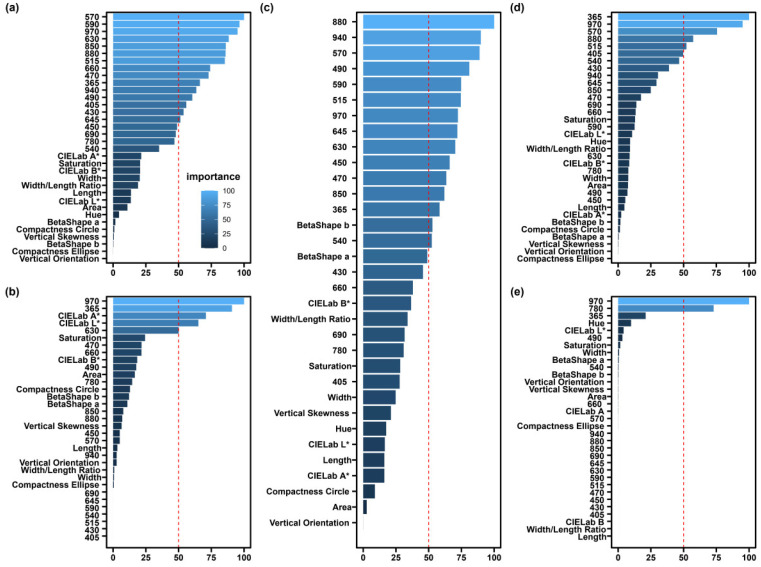
Importance score of morphological and spectral features for (**a**) MLR algorithm, (**b**) XGBoost algorithm, (**c**) SEL algorithm, (**d**) SVM algorithm, and (**e**) GLMNET algorithm. The Y-axis represents all of the features, and all of the feature explanations can be found in Appendix A. The X-axis indicates the feature importance scores. The blue bars indicate importance scores, with lighter colors resulting in higher scores and vice versa. It is important to note that the feature scores for each model are scaled to be in the range of 0–100. The symbol * indicate the color features.

**Figure 8 sensors-22-07521-f008:**
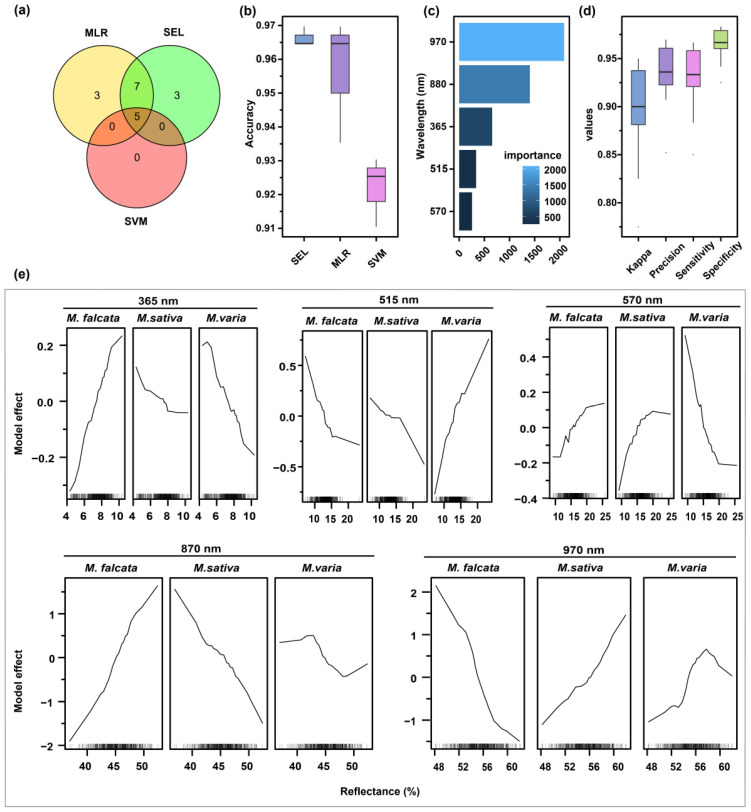
Selecting the optimal wavelengths and evaluating their performances. (**a**) The important feature intersection of SEL, SVM, and MLR models was selected as the optimal wavelength. (**b**) The accuracy of the three models, SEL, SVM, and MLR, was compared using the optimal wavelength as input data. (**c**) SEL performances were evaluated in terms of kappa, accuracy, sensitivity, and specificity. (**d**) Output of optimal wavelength importance was ranked in SEL. (**e**) The model effect of optimal wavelengths was assessed in SEL.

**Table 1 sensors-22-07521-t001:** Seeds information of *Medicago*.

Species	Cultivar	Origin	Moisture Content (%)	Harvest Year	Storage Temperature (°C)
*M. falcata*	Tenggeli	China	6.26	2020	25
*M. varia*	Caoyuan No.3	China	5.40	2020	25
*M. sativa*	Zhongmu No.1	China	5.12	2019	25

**Table 2 sensors-22-07521-t002:** Random-research details of partial hyperparameters of basic models and meta-learner.

Level	Models	Optimization Range of Partial Hyperparameters	Tuned Hyperparameters
Basic models	SVM	kernel = [‘polynomial’, ‘linear’, ‘sigmoid’, ‘radial’]; cost = [0:5]	kernel = ‘polynomial’; cost = 2.746
	GLMNET	α = [0:1]	α = 0.346
	XGBoost	booster = [‘gbtree’, ‘gbliner’, ‘dart’]; η = [0:0.5]; nrounds = [1:20, tags = ‘budget’]	booster = ‘gblinear’; η = 0.408; nrounds = 11
Meta-learner	XGBoost	Booster = [‘gbtree’, ‘gbliner’, ‘dart’]; η = [0:0.5]; nrounds = [1:20, tags = ‘budget’]	Booster = ‘gblinear’; η = 0.424; nrounds = 5

**Table 3 sensors-22-07521-t003:** Confusion matrix and metric for classification of SEL based on spectral data.

	Prediction	Reference			Total
		*M. falcata*	*M. sativa*	*M. varia*	
Training	*M. falcata*	140	0	0	-
(n = 140)	*M. sativa*	0	140	0	-
	*M. varia*	0	0	140	-
	Accuracy	1.00	1.00	1.00	1.00
	Sensitivity	1.00	1.00	1.00	1.00
	Specificity	1.00	1.00	1.00	1.00
	Precision	1.00	1.00	1.00	1.00
Testing	*M. falcata*	59	0	0	59
(n = 60)	*M. sativa*	0	60	0	0
	*M. varia*	1	0	60	1
	Accuracy	0.99	1.00	1.00	1.00
	Sensitivity	0.98	1.00	1.00	0.99
	Specificity	1.00	1.00	0.99	1.00
	Precision	1.00	1.00	0.98	0.99
CV	Accuracy	0.99 ± 0.01	1.00 ± 0.00	0.98 ± 0.02	0.98 ± 0.02
(n = 140)	Sensitivity	0.98 ± 0.02	1.00 ± 0.00	1.00 ± 0.00	0.98 ± 0.02
	Specificity	1.00 ± 0.00	1.00 ± 0.00	0.98 ± 0.02	0.99 ± 0.01
	Precision	1.00 ± 0.00	1.00 ± 0.00	0.98 ± 0.02	0.98 ± 0.02

Note: CV: cross-validation.

**Table 4 sensors-22-07521-t004:** Confusion matrix and metric for classification of SEL based on combination of spectral and morphological data.

	Prediction	Reference			Total
		*M. falcata*	*M. sativa*	*M. varia*	
Training	*M. falcata*	140	0	0	-
(n = 140)	*M. sativa*	0	137	2	-
	*M. varia*	0	3	138	-
	Accuracy	1.00	0.99	0.99	0.99
	Sensitivity	1.00	0.98	0.99	0.99
	Specificity	1.00	0.99	0.99	0.99
	Precision	1.00	0.99	0.98	0.99
Testing	*M. falcata*	59	0	1	-
(n = 60)	*M. sativa*	1	55	8	-
	*M. varia*	0	5	51	-
	Accuracy	0.99	0.92	0.90	0.94
	Sensitivity	0.98	0.92	0.85	0.92
	Specificity	0.99	0.93	0.96	0.96
	Precision	0.98	0.92	0.91	0.92
CV	Accuracy	0.99 ± 0.01	0.93 ± 0.05	0.92 ± 0.05	0.91 ± 0.05
(n = 140)	Sensitivity	0.99 ± 0.02	0.9 ± 0.11	0.90 ± 0.10	0.91 ± 0.05
	Specificity	0.99 ± 0.01	0.96 ± 0.04	0.95 ± 0.04	0.96 ± 0.02
	Precision	0.97 ± 0.02	0.93 ± 0.07	0.90 ± 0.07	0.92 ± 0.05

Note: CV: cross-validation.

## Data Availability

Not applicable.

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
