# Peer review of "Single Seed Identification in Three Medicago Species via Multispectral Imaging Combined with Stacking Ensemble Learning"

_sensors, 2022, doi:10.3390/s22197521_

Round 1

Reviewer 1 Report

The authors show an application case to detect through MSI analysis three kinds of seeds. This is interesting but is restricted to the studied seeds. The method could be generalized to other items but this is not explored.

I have some minor questions.

1. I would suggest to add seed images in the main article, not on online content. Now I was not able to see it so I don't know how the seeds looks like.

2. Line 196 to 200, equation texy shuld be all in cursive

3. Line 201, "Where" shuld be in lowercase and Line 200 should have a comma after the equation. Check all text equations.

4. Equation (6 to 8) references are not aligned.

5. XGBoost feature score should be explained in the text and in some equation form because is a black content for the reader that doesn't know this python library. There is some CIELab calculation but the used illuminant is not commented. Please comment it and also all the Figure 6 parameters should be explained. For example 570 Fig 6 a, What is the meaning?

This reviewer thinks this contribution should be published with this minor changes updated.

Reviewer 2 Report

Overall, this is a good piece of work that provides new ideas and new solutions for AI-assisted seed identification. The work is well conceptualized and executed. The presentation and writing are all of the high quality. 

Some minor points for authors to consider: 

1. The testing of 30% of the seeds as validation is reasonable and acceptable. It would be ideal to put this modeling into real-life blind testing to further prove its robustness and applicability. Testing an artificial mixed population will also be interesting. 

2. Just by the results of Figure 5, spectral data alone gives more accurate identification than using the combination of spectra and morphological features. I hope that the authors can further discuss this possibility. 

3. Please double-check the names of the three species for Figure 3. It seems that there are two M. sativa. 

Reviewer 3 Report

The paper is very well written. However, lines 142-152 may be explaine with an example for more clarity. Similarly Eqations 6,7 and 8 can be illustrated with examples for benefitof the readers.
